# Transcriptome Analysis of *Clinopodium chinense* (Benth.) O. Kuntze and Identification of Genes Involved in Triterpenoid Saponin Biosynthesis

**DOI:** 10.3390/ijms20112643

**Published:** 2019-05-29

**Authors:** Yuanyuan Shi, Shengxiang Zhang, Daiyin Peng, Chenkai Wang, Derui Zhao, Kelong Ma, Jiawen Wu, Luqi Huang

**Affiliations:** 1Anhui University of Chinese Medicine and Anhui Academy of Chinese Medicine, Hefei 230038, China; 15256513913@163.com (Y.S.); m13856034553@163.com (S.Z.); pengdy@ahtcm.edu.cn (D.P.); 15256912878@163.com (C.W.); m15755182557@163.com (D.Z.); makelong210@126.com (K.M.); 2Key Laboratory of Xin’an Medicine, Ministry of Education, Anhui University of Chinese Medicine, Hefei 230038, China; 3Synergetic Innovation Center of Anhui Authentic Chinese Medicine Quality Improvement, Hefei 230012, China; 4Clinical College of Integrated Traditional Chinese and Western Medicine, Anhui University of Chinese Medicine, Hefei 230012, China; 5State Key Laboratory Breeding Base of Dao-di Herbs, National Resource Center for Chinese Materia Medica, China Academy of Chinese Medical Sciences, Beijing 100700, China

**Keywords:** *Clinopodium chinense* (Benth.) O. Kuntze, differentially expressed genes, RNA sequencing, transcriptome, triterpenoid saponin biosynthesis

## Abstract

*Clinopodium chinense* (Benth.) O. Kuntze (*C. chinense*) is an important herb in traditional Chinese medicine. Triterpenoid saponins are a major class of active compounds in *C. chinense* with broad pharmacological activities and hemostatic, antitumor, and anti-hyperglycemic effects. To identify genes involved in triterpenoid saponin biosynthesis, transcriptomic analyses of leaves, stems, and roots from *C. chinense* were performed. A total of 135,968 unigenes were obtained by assembling the leaf, stem, and root transcripts, of which 102,154 were annotated in public databases. Differentially expressed genes were determined based on expression profile analysis and analyzed for differential expression of unique genes related to triterpenoid saponin biosynthesis. Multiple unigenes encoding crucial enzymes or transcription factors involved in triterpenoid saponin synthesis were identified and analyzed. The expression levels of unigenes encoding enzymes were experimentally validated using quantitative real-time PCR. This study greatly broadens the public transcriptome database for this species and provides a valuable resource for identifying candidate genes involved in the biosynthesis of triterpenoid saponins and other secondary metabolites.

## 1. Introduction

The species *Clinopodium chinense* (Benth.) O. Kuntze from the genus Clinopodium of the Lamiaceae family is recorded in the Chinese pharmacopoeia [1]. The aerial parts of *C. chinense*, as well as *Clinopodium polycephalum*, known as duanxueliu in China, are used as a traditional folk medicine for treating diseases such as hematuria, influenza, and allergic dermatitis [2]. Previous studies of the chemical constituents of *C. chinense* have indicated that triterpenoid saponins are the major chemical components of *C. chinense* [3]. Triterpenoid saponins exert important pharmacological effects including hemostatic [4], antitumor [5], and anti-hyperglycemic [6] activities. However, it is difficult to extract triterpenoid saponins in sufficient quantities from natural sources. Triterpenoid saponin biosynthetic pathways have not been well characterized. Genome sequencing and transcriptome profiling studies of species such as *C. chinense* have the potential to significantly improve understanding of these pathways.

Triterpenoid saponins are a class of structurally diverse specialized metabolites in plants [7] and marine invertebrates including sea cucumbers [8] and sponges [9]. Their biosynthetic routes include the isoprenoid pathway, and isopentenyl pyrophosphate (IPP) is the precursor of all isoprenoids [10]. IPP is synthesized via the mevalonate (MVA) or 2-C-methyl-D-erythritol-4-phosphate (MEP) pathway [10]. Triterpenoid saponin biosynthesis can be summarized in three main stages: first, IPP is converted to farnesyl pyrophosphate (FPP) by geranyl-diphosphate synthase (GPPS) and farnesyl-diphosphate synthase (FPPS) [11]. Second, 2,3-oxidosqualene is cyclized by 2, 3-oxidosqualene cyclases (e.g., beta-amyrin synthase (β-AS) and lupeol synthase (LS)) to form diverse compounds (e.g., beta-amyrin and lupeol) [12]. Finally, the formation of various triterpenoid saponins is catalyzed by cytochrome P450-dependent monooxygenases (PDMO) and glucosyltransferases (GTs) [13,14].

Recently, transcriptome analysis has become an effective approach to identify the biosynthesis of secondary metabolites and determine the functions of genes in plants. RNA sequencing (RNA-seq) has been widely used to screen functional genes and accurately quantify gene expression without a reference genome [15,16]. Several secondary metabolite biosynthesis pathways in medicinal plants have been analyzed, including terpenoid biosynthesis in *Artemisia argyi* [17], and triterpenoid saponin biosynthesis in *Anemone flaccida* [18] and *Gleditsia sinensis* [19].

In this study, we conducted a comprehensive transcriptome profile analysis of *C. chinense*, and identified numerous genes related to triterpenoid saponin biosynthesis. These transcriptome data provided new insights to guide further studies on this species.

## 2. Results

### 2.1. Total Saponin Content in C. chinense Samples

We extracted total saponins from the dried leaves, stems, and roots of *C. chinense*. Total saponin content was higher in the aerial parts of *C. chinense* (leaves, 0.157%; stems, 0.155%), but lower in roots (0.118%) (Appendix A).

### 2.2. Sequencing and de novo Assembly

Samples from C. chinense were sequenced using the BGISEQ-500 platform. After quality filtering, 30.51 Gb of clean reads were generated using an average Q30 of 90.32% (sequencing error rate < 1%), and 135,968 unigenes were obtained using the Trinity and TGI clustering tool (TGICL). The number of unigenes in leaf, stem, and root tissues were 64,540, 108,624, and 70,844, respectively. The N50 value was 1890 bp and the average length of the unigenes in C. chinense was 1195 bp; 72,498 (53.32%) of unigenes exceeded 500 bp, and 66,102 unigenes (48.61%) exceeded 1000 bp (Appendix A).

### 2.3. Unigene Functional Annotation and Overview of Unigene Expression

Among the 135,968 unigenes, 102,154 genes (75.13%) were annotated in public databases, including 70.86%, 51.42%, 53.76%, 55.01%, 56.03%, 53.73%, and 37.67% in NCBI nonredundant protein sequences (NR), NCBI nucleotide sequences (NT), a manually annotated and reviewed protein sequence database (SwissProt), Kyoto Encyclopedia of Genes and Genomes (KEGG), clusters of euKaryotic Orthologous Groups (KOG), Pfam, and gene ontology (GO), respectively (Table 1). According to Venn diagram analysis, 49,785 (36.62%) unigenes were co-annotated in five databases (Appendix A). Additionally, 96,353 unigenes were annotated in the NR database. Just over half (55.48%) of the annotated unigenes were mapped to Sesamum indicum, 20.10% were mapped to Erythranthe guttata, 19.76% were mapped to Dorcoceras hygrometricum, and 4.65% were mapped to others (Appendix A).

Moreover, we conducted an annotation of unigenes in C. chinense using the MAPMAN software. The most unigenes were enriched in the categories of “protein”, “RNA”, “signaling”, “miscellaneous function (misc)”, “transport”, and “stress” (Appendix A). Based on the unigenes with fragments per kilobase of transcript per million mapped reads (FPKM) > 1, an overview of metabolic and secondary metabolic pathways was developed using MAPMAN analysis. In the metabolic pathway, the most unigenes were mapped to “lipids metabolism” and “secondary metabolism”. In secondary metabolic pathways, the most unigenes were enriched in the “phenlypropanoids”, “flavonoids”, and “lignin and lignans” pathways (Appendix A).

The annotated unigenes were described using GO terms which were divided into three classes: biological process, cellular component, and molecular function; 51,220 unigenes were matched with one or more GO terms comprising of 55 subcategories. We focused on the biological process and molecular function categories in this study. The most abundant categories under molecular function were cellular process, metabolic process, and transporter activity (24,757, 24,210, and 2761 unigenes, respectively). The most abundant categories under biological processes were cellular process, metabolic process, and biological regulation (17,210, 16,877, and 4937 unigenes, respectively) (Appendix A).

The expression values of transcripts in each tissue were calculated based on FPKM > 1. The numbers of expressed unigenes were 46,136, 61,118, and 50,304 in leaf, stem, and root tissues, respectively (Figure 1A). We observed that the overall expression level of unigenes in roots was lower than in leaves or stems (Figure 1B).

### 2.4. Identification of Candidate Genes Involved in Triterpenoid Saponin Biosynthesis by KEGG Pathway Analysis

To determine the main biological processes in C. chinense, 74,791 unigenes were annotated in the KEGG database; these were classified into five categories (cellular process, genetic information processing, metabolism, organismal systems, and environmental information processing) (Appendix A) and distributed to 136 KEGG pathways (Appendix A). Seven pathways were assigned to the “metabolism of terpenoids and polyketides” subcategory, and the largest numbers of unigenes were associated with terpenoid backbone biosynthesis (Figure 2A). The biosynthesis of other secondary metabolites included 14 pathways, of which the unigenes were most enriched in phenylpropanoid biosynthesis (Figure 2B).

We annotated 708 unigenes involved in “terpenoid backbone biosynthesis” (KO00900), “sesquiterpenoid and triterpenoid biosynthesis” (KO00909) and “steroid biosynthesis” (KO00100) based on the KEGG database (Appendix A). Based on KEGG pathway analysis, we developed a model to summarize the biological pathways involved in triterpenoid saponin biosynthesis (Appendix A). Triterpenoid saponins are synthesized by the MVA pathway in cytoplasm and mitochondria or the MEP pathway in plastids. Moreover, IPP and Dimethylallyl diphosphate (DMAPP) are the precursors of all isoprenoids, including monoterpenoids, sesquiterpenoids, diterpenoids, triterpenoid saponins, steroids, and carotenoids. A total of 129 unigenes were identified as crucial for encoding seven key enzymes involved in triterpenoid saponin biosynthesis, including 3-hydroxy-3-methylglutaryl CoA reductase (HMGR) (15 unigenes), 1-deoxy-D-xylulose-5-phosphate synthase (DXS) (22 unigenes), 1-deoxy-D-xylulose-5-phosphate reductoisomerase (DXR) (10 unigenes), squalene synthase (SS) (26 unigenes), squalene monooxygenase (SM) (18 unigenes), β-AS (32 unigenes), and LS (6 unigenes) (Table 2) (Figure 3). The enzymes most closely associated with triterpenoid saponin biosynthesis were PDMOs and GTs. Three hundred twelve cytochrome P450s and 84 GTs were annotated in this study (Appendix A). The biosynthesis of triterpenoids was outlined based on genes encoding enzymes with FPKM > 1

As previously described, β-AS catalyzes the cyclization of 2,3-oxidosqualene to form triterpene skeletons, a critical branching point for phytosterol and triterpenoid biosynthesis. Six unigenes were confirmed to encode β-AS in this study by aligning their amino acids to the NCBI BLAST database (Appendix A). The alignment of six β-AS amino acid sequences showed that their sequence identity was 81.52%, and β-AS contained a characteristic region (MWCYCR) and a well-conserved binding site (DCTAE) (Figure 4). We chose three unigenes (i.e., CL2196. Contig2, CL5601. Contig1 and CL17709. Contig1) to construct 3D structural models based on the crystal structure of human OSC (PDB ID: 1w6j.1.A [20]) using the SWISS-MODEL (https://swissmodel.expasy.org/) (access on 27 January 2019) and PyMOL software. These β-AS models all contained abundant α-helices with “MWCYCR” and “DCTAE” motifs (Figure 4).

### 2.5. Differentially Expressed Gene Analysis in Leaf vs. Root, and Stem vs. Root Tissue

Differentially expressed genes (DEGs) within stem, leaf, and root tissues were screened using Poisson distribution methods with the parameters fold change (FC) ≥ 2.00 and false discovery rate (FDR) ≤ 0.001. Based on a Poisson distribution, 3372 unigenes showed expression in leaf tissues, 3287 unigenes showed expression in stem tissues, and 70,284 shared unigenes were identified in each of the three tissues (Figure 5A). Substantial transcription differences were observed in pairwise comparisons between different tissues. Forty-five-thousand-nine-hundred-and-sixty-one DEGs were commonly expressed in leaf and root tissue, while 14,951 DEGs were upregulated and 31,010 were downregulated in the leaf compared with the root. Comparison of stem and root tissue resulted in 33,110 DEGs, of which 12,570 were upregulated and 20,540 were downregulated in the stem compared with the root (Figure 5B).

Using KEGG enrichment, 33,088 DEGs were identified in leaf versus root, and 23,782 DEGs identified in stem versus root were mapped to 138 pathways, which were mainly enriched in “metabolic pathways”, “biosynthesis of secondary metabolites”, and “plantpathogen interaction” (Figure 6). Moreover, we identified 199 DEGs involved in terpenoid and polyketide metabolism, including 69 upregulated DEGs derived from leaf versus root and 36 upregulated DEGs derived from stem versus root (Figure 7).

GO enrichment analysis showed that 25,024 DEGs derived from leaf versus root analysis in the “biological process” category were mainly mapped to “photosynthesis”, “cell wall organization or biogenesis”, “carbohydrate metabolic process”, and “cellular polysaccharide metabolic process” (Appendix A). “Drug catabolic process”, “hydrogen peroxide metabolic process”, “photosynthesis”, and “hydrogen peroxide catabolic process” were the major enriched GO terms of DEGs derived from stem versus root analysis (Appendix A). Furthermore, 40,108 DEGs in molecular function were mainly assigned to “oxidoreductase activity” in leaf versus root and stem versus root analyses (Appendix A). In general, a p-value for each term for which FDR ≤ 0.01 was defined as significant enrichment.

### 2.6. Identification of Candidate Genes Involved in Hormone Biosynthesis by MAPMAN Analysis

Plant hormones play an important part in all stages of plant growth, especially in regulating secondary metabolites. Using the MAPMAN software, we identified 10 unigenes involved in gibberellin (GA) biosynthesis, 27 unigenes involved in abscisic acid (ABA) biosynthesis, and 60 unigenes involved in jasmonate (JA) biosynthesis (Appendix A). Furthermore, we identified 14 upregulated unigenes in the leaf versus root comparison and 36 upregulated unigenes in the stem versus root comparison (Table 3).

### 2.7. Detection of Transcription Factor Families

Transcription factor (TF) families participate in a wide variety of biological processes in plants and have important roles in regulating the activity of triterpenoid saponin biosynthesis and other secondary metabolic processes. A total of 4381 unigenes encoding TFs were identified in the C. chinense transcriptome database and classified into 59 different TF families, including 752 upregulated unigenes in the leaf versus root comparison and 561 upregulated unigenes in the stem versus root comparison (Table 4). We concluded that the MYB family (572 unigenes) accounted for the largest proportion of TF families, followed by MYB-related (433 unigenes), AP2-EREBP (345 unigenes), bHLH (282 unigenes), WRKY (277 unigenes), NAC (206 unigenes), GRAS (168 unigenes), and C3H (160 unigenes). Furthermore, we confirmed that the MYB (nine unigenes), MYB-related (nine unigenes), and FHA (five unigenes) TF families were involved in metabolism of terpenoids and polyketides, and that 12 TF families participated in biosynthesis of other secondary metabolites (Figure 8).

### 2.8. Validation of Unigenes and Gene Expression Profiling Using qRT-PCR

We conducted quantitative real-time PCR (qRT-PCR) experiments to validate the expression patterns of the DXS, DXR, HDS, PMK, IDI and FPPS genes. Relative expression patterns of DXS, DXR, HDS, and FPPS showed greater expression in leaf tissue, whereas PMK showed greater expression in stem tissue and IDI showed greater expression in root tissue (Figure 9).

## 3. Discussion

Although *C. chinense* exhibits important pharmacological activities owing to its triterpenoid saponins, biosynthesis of triterpenoid saponins has not been characterized. Our study aimed to identify the candidate genes that encode key enzymes related to triterpenoid saponin biosynthesis and other secondary metabolic pathways. In this study, the transcriptomes of *C. chinense* derived from three tissues were acquired using the BGISEQ-500 technique, resulting in 30.51 Gb of clean reads that were then assembled into 135,968 unigenes with an average length of 1195 bp. Among these unigenes, 102,154 (75.13%) were mapped to seven public databases. Compared with other medicinal plant transcriptome databases, the average length and N50 values of unigenes in *C. chinense* (average length = 1195 bp; N50 = 1890 bp) were longer than those in *Artemisia argyi* (average length = 926 bp; N50 = 1456 bp) [17], *Oroxylum indicum* (average length = 1080 bp; N50 = 1783 bp) [21], and *Asarum heterotropoides* (average length = 611 bp; N50 = 507.36 bp) [22]; these results demonstrated that our transcriptome database was of high quality. In particular, the sequence size distribution was homogeneous and 10,942 (48.61%) unigenes were longer than 1000 bp, indicating successful generation of transcriptional data.

Based on GO term enrichment, we focused mainly on the categories of biological process and molecular function. The most enriched ontologies were photosynthesis (leaf versus root) and drug catabolic process (stem versus root) in biological processes, while the abundant subcategory of molecular function was oxidoreductase activity (leaf versus root and stem versus root), which might be significant for the activity of cytochrome oxidase P450 in the triterpenoid saponin biosynthesis.

KEGG pathway enrichment analysis led to the identification of 708 unigenes relevant to triterpenoid saponin biosynthesis. Among these, we found that unigenes encoding DXS (CL2016. Contig 1, 3, 5, 7, and 8), and DXR (CL10703. Contig 1, 2, 3, and 6) were more highly expressed in leaves compared with other tissues. Studies showed that DXS and DXR were the key enzymes in the MEP pathway [23,24]. In previous studies, overexpression of DXS gene in kiwifruit directly resulted in a significant increase in monoterpenoid levels in transgenic tobacco leaves [25], and overexpression of DXR in *Artemisia annua* L. led to an approximately 2-fold increase in artemisinin production by greatly influencing the biosynthesis of terpenoids [26]. The expression levels of unigenes encoding DXS (CL2016. Contig 3), DXR (CL10703. Contig 3), HDS (CL10042.Contig3), PMK (CL8457.Contig10), IDI (CL10625.Contig5) and FPPS (CL7015.Contig4) were determined by qRT-PCR to verify our transcriptional data were authentic and reliable. Characterization of these unigenes contributed to our understanding of the molecular mechanisms underlying triterpenoid saponin biosynthesis.

Studies have shown that triterpenoid saponin possess two conformations (“chair–chair–chair” and “chair-boat-chair”), which form the precursors of steroids and triterpenoids cyclized by 2,3-oxidosqualene cyclases [12]. The main active ingredients of *C. chinense* are oleanane-type triterpenoid saponins [3]. β-AS is thought to catalyze the cyclization of 2, 3-oxidosqualene to form β-amyrin, the basic triterpene backbone of oleanane-type saponins [11]. This step is a critical branching point for phytosterol and triterpenoid biosynthesis [27,28]. Six β-AS unigenes were identified in *C. chinense* datasets, and the alignment of six β-AS unigenes suggested that β-AS contains a highly conserved binding site and characteristic motif (Figure 4). These results were consistent with β-AS in other plants [28,29]. Previous studies have shown that the “DCTAE” motif is the initiation site for the polycyclization reaction. The Asp residue in this motif releases protons to trigger the cyclization reaction in the conversion of 2,3-oxidosqualene to β-amyrin [28,30]. The “MWCYCR” motif is a characteristic motif of β-AS. the “W” residue controls the formation of β-amyrin by stabilization of an oleanyl cation, and the “Y” residue participates in forming pentacyclic triterpenes [29]. Moreover, the rich helix suggests that β-AS is a membrane-related protein [20]. Despite the sequence diversity of these genes, the protein 3D structures were conserved and had similar functions.

The overall expression level of unigenes and content of total saponins in root tissue was lower than that in leaf and stem tissues. This result suggested that the aerial parts of *C. chinense* might contain effective medicinal compounds. Based on DEG analysis, 69 DEGs involved in terpenoid backbone biosynthesis were upregulated in leaf versus root tissue, and 36 DEGs were upregulated in stem versus root tissue. The observation that these upregulated DEGs control the biosynthesis of the terpenoid backbone in leaf and stem tissue further confirmed that the aerial parts of *C. chinense* possess important medicinal value.

Many TFs are difficult to detect owing to their low expression levels; however, they are very important because small increases in expression levels of TFs can have drastic effects [31]. In our study, a total of 4381 candidate TFs were assigned to 59 TF families. These TFs might be crucial for plant metabolism and regulation. MYB TFs are crucial for biosynthesis of the terpenoid backbone. A previous investigation indicated that the overexpression of a MYB TF in tomato can upregulate the terpenoid metabolism [32]. The 572 candidate MYB TFs discovered in our dataset included 100 upregulated TFs in leaf versus root comparisons and 75 upregulated TFs in stem versus root comparisons (Table 3). Previous studies also showed that the overexpression of WRKY in transiently transformed *C. blinii* resulted in improved total saponin content [33]. A total of 277 WRKY TFs were identified in this study, of which 76 upregulated TFs were identified in the leaf versus root comparisons and 29 upregulated TFs were identified in stem versus root comparisons. Specially regulated TFs might be responsible for modulating the content of triterpenoid saponins in *C. chinense*.

Plant hormones are plant-specific key signaling molecules that respond to various stimuli and are involved in the synthesis of secondary metabolites [34]. JA can upregulate the expression level of the squalene synthase (BFSS1) gene, stimulate the accumulation of β-AS mRNA and increase the content of bupleurum saponins [35]. Additionally, the synthesis of sesquiterpene and monoterpene were promoted in GAs-mediated grapevine [36]. In the present study, we found 60 genes involved in JA biosynthesis, including eight and 24 upregulated genes in leaf vs. root and stem vs. root comparisons, respectively. Ten genes participated in GA biosynthesis, of which only one upregulated gene was identified in the stem versus root comparisons, which may indicate the importance of the biosynthesis of GA.

## 4. Materials and Methods 

### 4.1. Sample Preparation for Transcriptome Sequencing and RNA Isolation

A series of whole *C. chinense* plants were collected from the herb garden of Anhui University of Chinese Medicine and were authenticated by Professor Qingshan Yang (Anhui University of Chinese Medicine). The plants were cleaned with ultrapure water, separated into three parts (leaves, stems, and roots), then frozen in liquid nitrogen immediately and stored at −80 °C to preserve RNA. Total RNA was extracted from three replicates, which were then pooled together using RNA Plant Plus Reagent (Tiangen, Beijing, China) according to the manufacturer’s instructions. The concentration of the isolated RNA, the 28S/18S ratio, and RNA integrity number were verified using an RNA Nano 6000 Assay Kit with the Agilent Bioanalyzer 2100 system (Agilent, CA, USA) (Appendix A).

### 4.2. Determining Total Saponins Content

Dried *C. chinense* samples from leaves, stems, and roots were used for separation of total saponins using a similar method to that previously reported [37,38]. Dried powder (0.1 g) from each sample was mixed with 50% carbinol and then subjected to ultrasonic extraction for 40 min (300 W, 40 kHz). The supernatant was then collected, dried by distillation, and dissolved in carbinol. Absorbance was measured using an ultraviolet spectrophotometer (Shimadzu Corporation, Japan). Clinopodiside A was used as a standard and the standard curve of the relationship between concentration and absorbance was constructed (Appendix A). The yield (%) of total saponins was calculated as Yield (%) = [saponin content of extraction (g)/*C. chinense* samples powder weight (g)] × 100%].

### 4.3. Library Construction and Sequencing

Messenger RNA was purified from total RNA by oligo (dT) magnetic beads. After purification, the mRNA was broken into 200–300 bp fragments using fragmentation buffer. First-strand cDNA was synthesized using the RNA fragments as templates. Second-strand cDNA was synthesized using dNTPs, RNase H, and DNA polymerase I. Short cDNA fragments were recovered and repaired, subjected to 3’ single adenylation, and ligated with sequencing adapters. The cDNA samples were subjected to PCR amplification to select the appropriate cDNA fragments. Each cDNA library was quantified and evaluated using an Agilent 2100 Bioanalyzer and ABI StepOnePlus Real-Time PCR System. The cDNA library was constructed using a BGISEQ-500 platform.

### 4.4. De novo Transcriptome Assembly and Unigene Functional Annotation

To ensure the accuracy of de novo assembly and subsequent analyses, the raw reads and low quality reads (above 50% of bases with Q-value ≤ 20), ambiguous reads, adaptor sequences, and duplication sequences were removed before assembly. Clean reads were assembled into contigs using Trinity software [39]. All transcripts were analyzed on the BGISEQ-500 platform [40]. The assembled transcripts were extended and clustered using the TGICL software [41]. The assembled transcripts were processed for further functional annotation and classification analysis.

Unigene functional annotation was achieved by mapping unigenes to five databases (NT, NR, KOG, KEGG, and SwissProt) using the software BLAST (version 2.2.23, E-value ≤ 1e-5) [42]. Morever, unigenes were mapped to metabolic and secondary metabolic pathways using MAPMAN (version 3.6.0) [43]. GO functional annotation was performed using Blast2GO (version 2.5.0, default parameters) [44] with NR annotations and Pfam annotations were performed using Hmmscan [45].

### 4.5. Analysis of Differentially Expressed Genes

The clean reads of each samples were mapped to unigenes using Bowtie2 (version 2.2.5) [46] software based on transcriptome assembly. To compare unigene expression levels between two tissues (leaf versus root tissue and stem versus root tissue), FC ≥ 2.00 and FDR ≤ 0.001 were considered to indicate significant differences in gene expression using the PoissonDis method [47]. DEGs were used for GO and KEGG enrichment analysis following the method described by Audic [46].

In the GO functional analysis, a hypergeometric test was applied for all DEGs mapped to terms in the GO database, in order to detect significantly enriched GO terms in DEGs compared with the whole transcriptome of *C. chinense*. The *p*-value was calculated as follows:$$p=1-\sum_{i=0}^{m-1}(Mi)(N-Mn-i)/(Nn)$$
where *N* and *n* represent the number of annotated unigenes with GO annotations and DEGs in *N*, respectively. *M* and m represent the annotated unigenes corresponding to certain GO terms and DEGs in *M*, respectively. The KEGG database was used to identify signal transduction or significantly enriched metabolic pathways compared with the transcriptome background. The *p*-value was calculated as described in the previous GO annotations analysis.

### 4.6. Identification of Transcription Factors

Open reading frames of each unigene detected with Getorf (parameter: -minsize150) [48] were aligned to TF protein domains in PlnTFDB (plant TF database) on the basis of BLASTX (E-value ≤ 1e-5) using Hmmsearch [44]. PlnTFDB was used to describe the properties of unigenes based on the characteristics of TFs.

### 4.7. qRT-PCR Analysis of Key Genes in Triterpenoid Saponin Biosynthesis

To validate the results of this de novo RNA-seq experiment, we chose 6 genes for qRT-PCR analysis using a QuantiNova SyBr Green PCR kit (Qiagen, Hilden, Germany) on a PIKOREAL 96 Real-Time Detection System (Thermo Scientific, Waltham, MA, USA). The Primer v5.0 software was used to design unigene-specific primers for qRT-PCR (Appendix A). Each reaction was performed in a final volume of 10 µL containing 5 µL of 2 × SYBR Green mixture, 1 µL of forward primer (10 µM), 1 µL of reverse primer (10 µM), 1 µL of cDNA, and 2 µL of RNase-free water. All reactions were performed under the following conditions: 95 °C for 1 min, 40 cycles of 95 °C for 20 s, and 60 °C for 1 min. To confirm the specificity of the amplicon for products, a melting curve was generated from 60 °C to 95 °C at the end of the PCR run. The relative expression level of each selected unigene was normalized to the actin gene (Unigene1915) and calculated using the 2^−ΔΔCT^ method [49].

## 5. Conclusions

In this study, we performed the transcriptome analysis of leaf, stem and root tissues in *C. chinense*, and identified numerous genes related to triterpenoid saponin biosynthesis using RNA-seq sequencing. A few genes encoding key enzymes were validated by qRT-PCR and the results were well in accordance with the expression data obtained by RNA-seq sequencing. This study will be useful to support our understanding of the mechanism of triterpenoid saponin biosynthesis in *C. chinense* at the molecular level. It could also greatly assist the research in molecular biology and mass production of triterpenoid saponins.

## Figures and Tables

**Figure 1 ijms-20-02643-f001:**
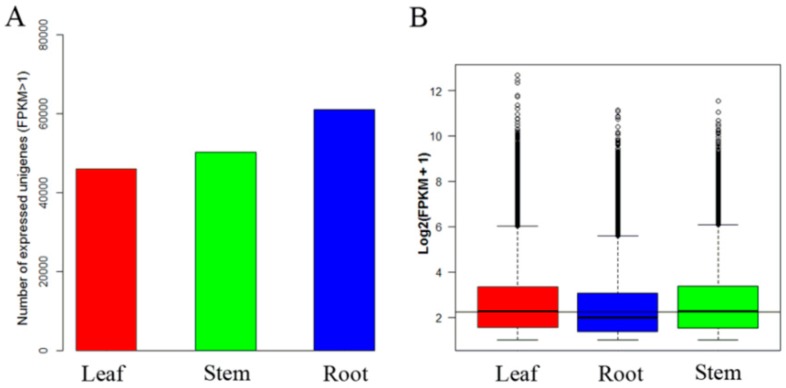
Overall expression profiles in C. chinense leaf, stem and root tissues. (**A**) Distribution of number of expressed unigenes (FPKM > 1) in tissues. (**B**) Boxplot of unigene expression in tissues. In (**B**), the x-axis represents tissue type and the y-axis represents log2 (FPKM + 1) values.

**Figure 2 ijms-20-02643-f002:**
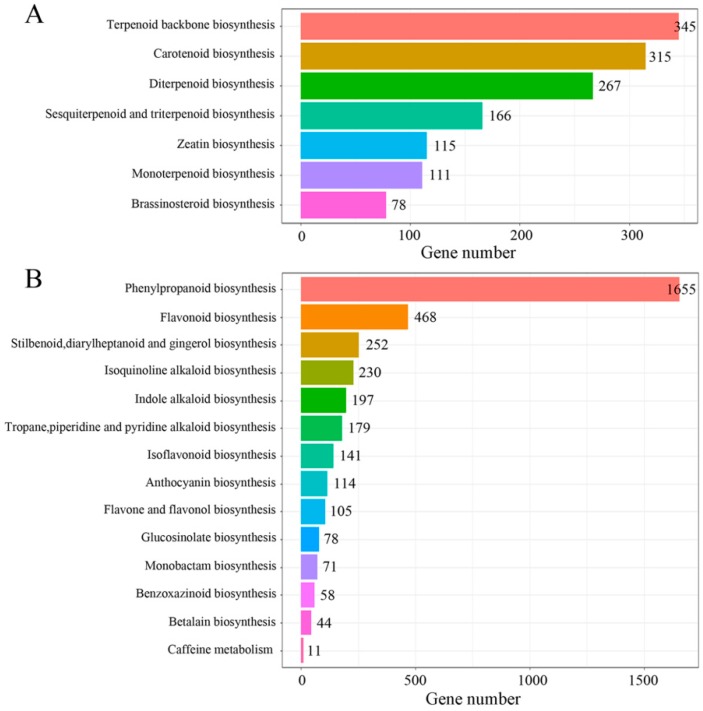
Kyoto Encyclopedia of Genes and Genomes (KEGG) annotation of C. chinense. (**A**) Classifications based on the metabolism of terpenoids and polyketides. (**B**) Classifications based on biosynthesis of other secondary metabolites.

**Figure 3 ijms-20-02643-f003:**
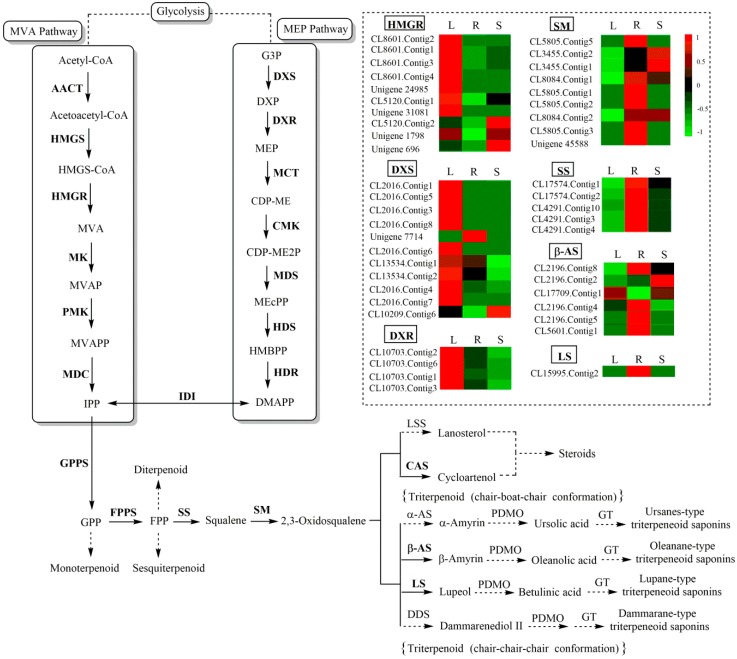
Pathway for triterpenoid saponin biosynthesis in C. chinense. The expression levels of unigenes encoding key enzymes from each process are shown using a heatmap. The columns L, R, and S correspond to leaf, root, and stem samples, respectively; red and green represent high and low expression levels, respectively.

**Figure 4 ijms-20-02643-f004:**
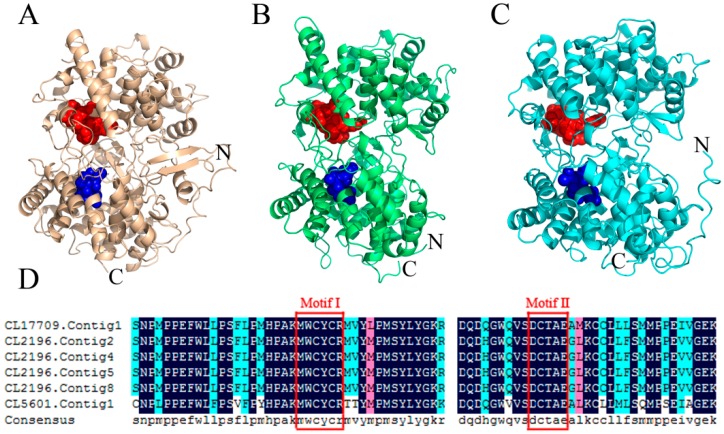
Spatial structure models of β-AS and the highly conserved motifs. (**A**–**C**) are the spatio-structural models of β-AS (CL2196.Contig2, CL5601.Contig1, and CL17709.Contig1; template: 1w6j.1.A; sequence identity: 43.10%, 39.88%, and 44.94%, respectively). The highly conserved MWCYCR (Motif I) and DCTAE (Motif II) residues are depicted as spheres in red and blue, respectively. (**D**) Alignment of β-AS amino acid sequences. Dark blue indicates identical amino acids, and red and lightskyblue indicate similar amino acids. Multiple sequence alignment was performed using the DNAMAN 6.0.3.99 software.

**Figure 5 ijms-20-02643-f005:**
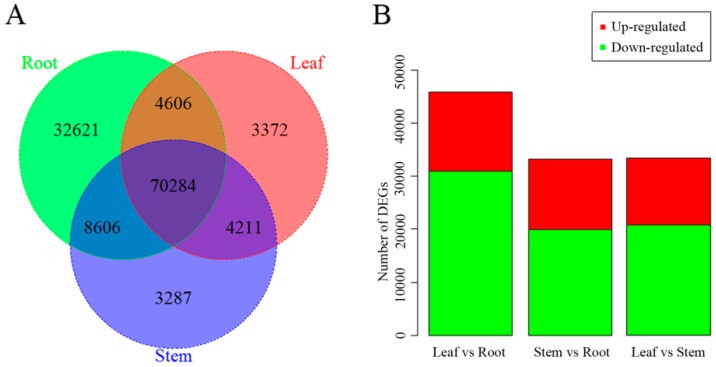
Expression of unigenes in C. chinense leaf, stem and root tissues. (**A**) Venn diagram of the number of unigenes expressed in stem, leaf and root tissues. (**B**) Number of upregulated and downregulated DEGs in leaf vs. root, stem vs. root and leaf vs. stem datasets.

**Figure 6 ijms-20-02643-f006:**
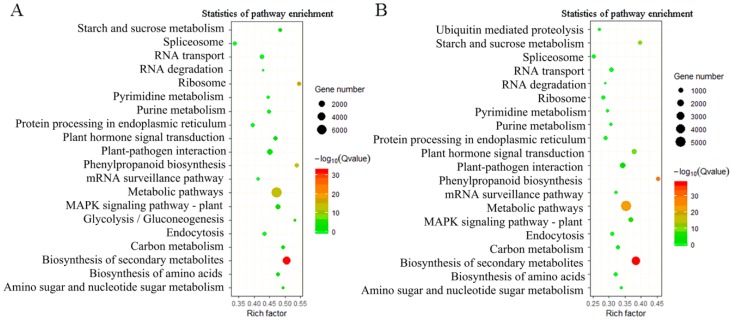
Enrichment of differentially expressed genes (DEGs) in KEGG pathway. (**A**) Significantly enriched pathways for DEGs in leaf versus root tissues. (**B**) Significantly enriched pathways for DEGs in stem versus root tissues. The x-axis represents the rich factor (the value of the enrichment factor, which is the quotient of the foreground value (number of DEGs) and background value (total gene amount)), and the y-axis represents KEGG pathways.

**Figure 7 ijms-20-02643-f007:**
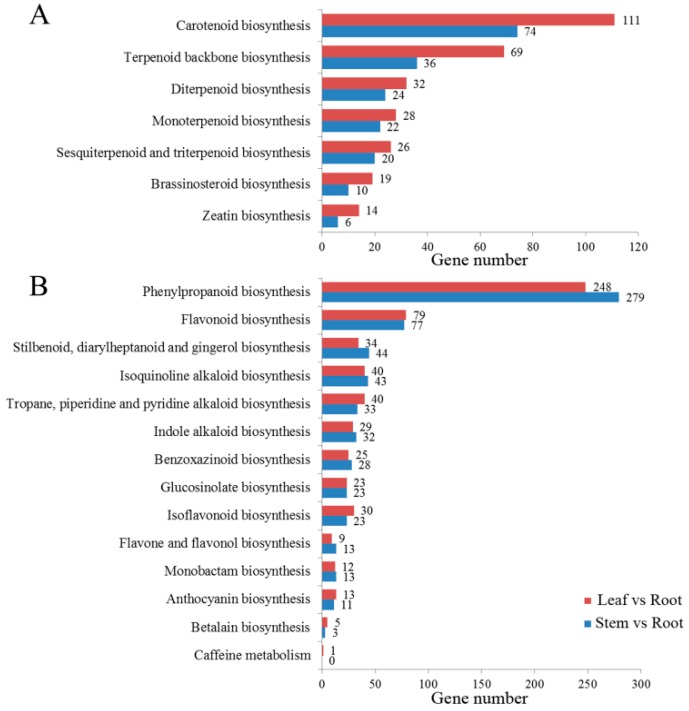
Number of DEGs in the most enriched pathways. (**A**) Number of upregulated DEGs in stem vs. root and leaf vs. root samples involved in terpenoid and polyketide metabolism. (**B**) Number of upregulated DEGs in stem vs. root and leaf vs. root samples involved in the biosynthesis of other secondary metabolites.

**Figure 8 ijms-20-02643-f008:**
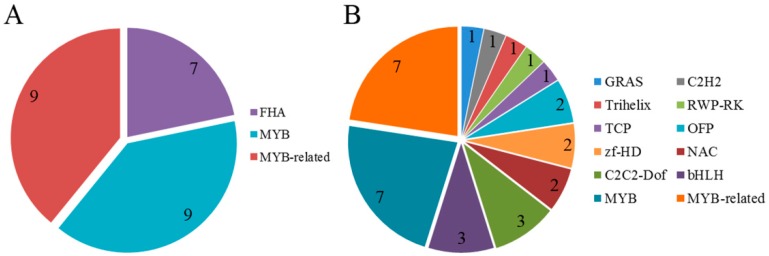
Transcription factor (TF) families in the metabolic pathway. (**A**) TF families in the metabolism of terpenoids and polyketides. (**B**) TF families in the biosynthesis of other secondary metabolites.

**Figure 9 ijms-20-02643-f009:**
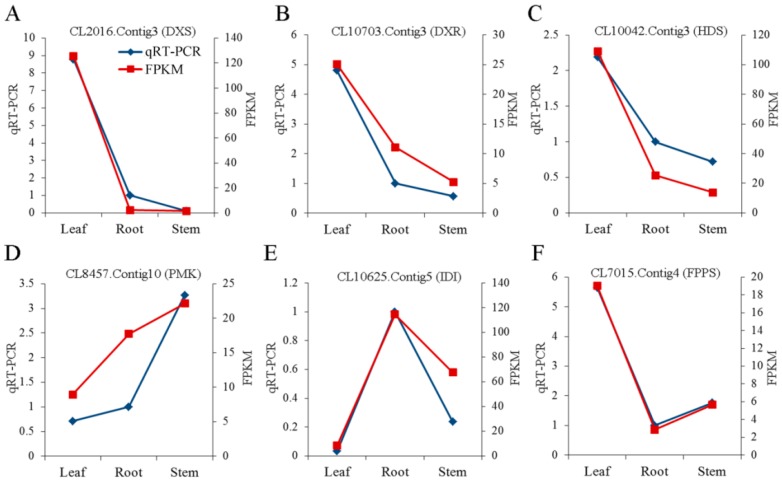
qRT-PCR analysis of unigenes encoding enzymes involved in triterpenoid saponin biosynthesis. Relative expression of (**A**) CL2016.Contig3 (DXS), (**B**) CL10703.Contig.3 (DXR), (**C**) CL10042.Contig3 (HDS), (**D**) CL8457.Contig10 (PMK), (**E**) CL10625.Contig5 (IDI), and (**F**) CL7015.Contig4 (FPPS) was analyzed by qRT-PCR using the actin gene (Unigene1915) as the reference gene for normalization. Red lines indicate the FPKM values of these unigenes, and blue lines indicate the qRT-PCR results.

**Table 1 ijms-20-02643-t001:** Summary statistics of annotations for C. chinense unigenes via seven public databases.

Database	Number Annotated	Annotated Unigene Ratio (%)
NR	96,353	70.86
NT	69,913	51.42
SwissProt	73,100	53.76
KOG	76,187	56.03
KEGG	74,791	55.01
Pfam	73,053	53.73
GO	51,220	37.67
Overall	102,154	75.13

**Table 2 ijms-20-02643-t002:** Number of unigenes encoding the key enzymes involved in the biosynthesis of triterpenoid saponins in C. chinense.

Abbreviation	EC number	Unigene Number	No. in Stems	No. in Roots	No. in Leaves
AACT	2.3.1.9	25	20	24	20
HMGS	2.3.3.10	8	4	8	4
HMGR	1.1.1.34	15	11	14	8
MK	2.7.1.36	4	2	4	2
PMK	2.7.4.2	12	8	11	10
MDC	4.1.1.33	17	10	13	9
DXS	2.2.1.7	22	17	19	18
DXR	1.1.1.267	10	8	10	9
MCT	2.7.7.60	6	6	6	6
CMK	2.7.1.148	3	3	3	3
MDS	4.6.1.12	3	2	3	2
HDS	1.17.7.1, 1.17.7.3	6	5	6	5
HDR	1.17.7.4	18	13	15	11
IDI	5.3.3.2	13	7	13	7
FPPS	2.5.1.1, 2.5.1.10	55	42	53	37
SS	2.5.1.21	26	20	26	19
SM	1.14.14.17	18	15	17	15
β-AS	5.4.99.39	32	28	28	27
LS	5.4.99.41	6	4	5	4
CAS	5.4.99.8	20	14	16	13

**Table 3 ijms-20-02643-t003:** Classification and number of hormone identified in the DEGs database for C. chinense.

Hormone	Number of Unigenes	Number of Upregulated Genes	
		Leaf vs. Root	Stem vs. Root
JA	60	8	24
GA	10	0	1
ABA	27	6	11
Total number	97	14	36

**Table 4 ijms-20-02643-t004:** Classification and number of TF families identified in the DEGs database of C. chinense.

TF Family	Number of Unigenes	Number of Upregulated Genes	
		Leaf vs. Root	Stem vs. Root
MYB	572	100	75
MYB-related	433	66	54
AP2-EREBP	345	19	21
bHLH	282	64	47
WRKY	277	76	29
NAC	206	32	13
GRAS	168	15	14
C3H	160	16	17
G2-like	132	30	17
C2H2	102	18	9
MADS	102	31	29
Trihelix	92	16	17
Tify	89	21	11
HSF	86	12	8
mTERF	78	21	17
C2C2-Dof	76	14	2
FAR1	71	14	10
RWP-RK	68	16	7
C2C2-GATA	64	9	6
ABI3VP1	62	4	11
ARF	61	0	14
SBP	60	7	21
Alfin-like	50	5	5
TAZ	49	14	11
TUB	47	0	2
bZIP	44	6	6
LOB	43	1	0
TCP	40	18	7
LIM	40	4	8
FHA	40	7	6
other	442	96	67
Total number	4381	752	561

## Data Availability

The RNA-seq datasets of three *C. chinense* tissues were deposited in the NCBI Sequence Read Archive (SRA) database (Accession: SRP166297).

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
