# Peer review of "Transcriptome Analysis of Clinopodium chinense (Benth.) O. Kuntze and Identification of Genes Involved in Triterpenoid Saponin Biosynthesis"

_ijms, 2019, doi:10.3390/ijms20112643_

Round 1
Reviewer 1 Report
The article concerns the investigation of all the transcriptomic materials involving in triterpene glycoside biosynthesis in Clinopodium chinense that is important in traditional Chinese medicine. The saponins are major active metabolites шт this plant and possess wide spectrum of biological activities including hemostatic, anti-tumor, and anti-hyperglycemic ones. To identify genes involved in triterpenoid saponin biosynthesis, full transcriptomic analyses of leaves, stems, and roots from C. chinense were performed. The differentially expressed genes (DEGs) were found by expression profile analysis in order to find unique genes related to triterpen glycoside biosynthesis. The expression level of unigenes encoding key enzymes was independently confirmed using real-time PCR procedure. The saponin contents in leaves, stems and roots was also determined for comparison. The authors have found specific genes related to specific squalencyclases, glycosyltranspherases and cytochromes responsible for different stages of saponin biosynthesis. These data correspond with the real presence of saponins in different part of the plant. The study provides an additional resource for finding the candidate genes responsible for biosynthesis of triterpenoid saponins.
The article is well written, the results are clearly described and the article may be published.
Nevertheless only one correction seems to be necessary (Page 2, Line 52). The phrase „Triterpenoid saponins are a class of structurally diverse specialized metabolites in plants [6]” should be replaces with „Triterpenoid saponins are a class of structurally diverse specialized metabolites in plants[6] and marine invertebrates including sea cucumbers [6a] and sponges [6b].” where [6a] = Mondol M.A.M., Shin, H.J., Rahman M.A., Islam M.T. Sea cucumber glycosides: chemical structures, producing species and important biological properties // Marine Drugs. 2017. V. 15, No. 10. 317; and [6b] = Kalinin V.I., Ivanchina N.V., Krasokhin V.B., Makarieva T.N., Stonik V.A. Glycosides from marine sponges (Porifera, Demospongiae): structures, taxonomical distribution, biological activities and biological roles // Marine Drugs. 2012. V. 10, No. 8. P. 1671–1710.
Hence, the article may be published after very minor corrections.
Author Response
Response to Reviewer 1 Comments
Point 1: Nevertheless only one correction seems to be necessary (Page 2, Line 52). The phrase “Triterpenoid saponins are a class of structurally diverse specialized metabolites in plants [6]” should be replaces with “Triterpenoid saponins are a class of structurally diverse specialized metabolites in plants[6] and marine invertebrates including sea cucumbers [6a] and sponges [6b].” where [6a] = Mondol M.A.M., Shin, H.J., Rahman M.A., Islam M.T. Sea cucumber glycosides: chemical structures, producing species and important biological properties // Marine Drugs. 2017. V. 15, No. 10. 317; and [6b] = Kalinin V.I., Ivanchina N.V., Krasokhin V.B., Makarieva T.N., Stonik V.A. Glycosides from marine sponges (Porifera, Demospongiae): structures, taxonomical distribution, biological activities and biological roles // Marine Drugs. 2012. V. 10, No. 8. P. 1671–1710.
Response 1: According to the reviewer’s suggestion, we have replaced ‘Triterpenoid saponins are a class of structurally diverse specialized metabolites in plants [6]’ with ‘Triterpenoid saponins are a class of structurally diverse specialized metabolites in plants [7] and in marine invertebrates including sea cucumbers [8] and sponges [9].’ (line 52-53).
Reviewer 2 Report
Transcriptome analysis of Clinopodium chinense (Benth.) O. Kuntze and identification of genes involved in triterpenoid saponin biosynthesis.
In this manuscript, authors described a transcriptome profile analysis of leaves, stems, and roots from Clinopodium chinense and identified several genes related to triterpenoid saponin biosynthesis.
The abstract should include information about the results of numerous genes related to triterpenoid saponin biosynthesis. This study represents most of the content in this paper.
The introduction seems to be appropriate respect the manuscript context, is easy to follow, clear.
The section of materials and methods is well described and detailed.
Results are clear but need some revision to describe the result appropriately.
The discussion seems to be well integrated with the results.
However, several aspects need some clarification and some more information is required.
1. Plant hormone and transcription factors play an important role in any biosynthesis pathways. Authors have to investigate the role of hormone biosynthesis or signalling genes in triterpenoid saponin biosynthesis.
2. The functional annotation analysis of DEGs: a detail description is essential.
3. An overview of metabolism and secondary metabolic pathways has to show by MAPMAN analysis.
4. Analysis of other related pathways will be helpful to make a conclusion.
5. Validation of more gene expression profiling is required.
6. The author should make a model of summary of some biological pathways involved in triterpenoid saponin biosynthesis.
Minor:
1. Is there any significant statistical difference about the total saponins content from leaves, stems and roots of Clinopodium chinense (Supplementary Figure S1)?
2. Line 249: fruit use two times
Author Response
Response to Reviewer 2 Comments
Major issues:
Point 1: Plant hormone and transcription factors play an important role in any biosynthesis pathways. Authors have to investigate the role of hormone biosynthesis or signalling genes in triterpenoid saponin biosynthesis.
Response: To address the reviewer suggestion, we have investigated the unigenes involved in hormone biosynthesis (line 213-220 and 318-325).
Point 2: The functional annotation analysis of DEGs: a detail description is essential.
Response: According to the reviewer’s comment, we have provided a detail description of the functional annotation analysis of DEGs (line 377-385).
Point 3: An overview of metabolism and secondary metabolic pathways has to show by MAPMAN analysis.
Response: According to the reviewer’s suggestion, we have included an overview of metabolic and secondary metabolic pathways produced by MAPMAN analysis (line 95-102 and 366-367) (Supplementary Fig. S4-S6).
Point 4: Analysis of other related pathways will be helpful to make a conclusion.
Response: According to the reviewer’s suggestion, we have analyzed the other related pathways in line 58-62 and 282-285 (Fig. 3).
Point 5: Validation of more gene expression profiling is required.
Response: According to the reviewer’s suggestion, we have supplemented more gene expression profiling (line 237-248 and 277-280) (Fig. 9).
Point 6: The author should make a model of summary of some biological pathways involved in triterpenoid saponin biosynthesis.
Response: We have developed a model of summarizing some of the biological pathways involved in triterpenoid saponin biosynthesis (Supplementary Fig. S12).
Minor issues:
Point 1: Is there any significant statistical difference about the total saponins content from leaves, stems and roots of Clinopodium chinense (Supplementary Figure S1)?
Response: In this study, the total saponin contents of leaves are similar to those of stems, whereas the total saponins contents of leaves or stems are about 1.3 times than those of roots. The leaves and stems are medicinal parts used in Traditional Chinese Medicine.
Point 2: Line 249: fruit use two times
Response: We have deleted the repeated word ‘fruit’ in line 274.
Round 2
Reviewer 2 Report
Authors have substantially improved the manuscript. I have some minor issues.
1. The statistical symbol should be in Supplementary Figure 1, if it is significant. Please write the figure legend properly.
2. Please write the figure legends properly for Supplementary Figure S4, S5, S6, S7, S8 and S12.
3. Based on KEGG pathway analysis, we developed a model to summarize the biological pathways involved in triterpenoid saponin biosynthesis (Supplementary Fig. S12).
Please explain it in a few lines.
Author Response
Minor issues:
Point 1: The statistical symbol should be in Supplementary Figure 1, if it is significant. Please write the figure legend properly.
Response: According to the reviewer’s suggestion, the statistical symbol and figure legend have been added to the Supplementary Figure S1.
Point 2: Please write the figure legends properly for Supplementary Figure S4, S5, S6, S7, S8 and S12.
Response: According to the reviewer’s suggestion, the figure legends have been added to Supplementary Figures S4, S5, S6, S7, S8, and S12.
Point 3: Based on KEGG pathway analysis, we developed a model to summarize the biological pathways involved in triterpenoid saponin biosynthesis (Supplementary Fig. S12).
Please explain it in a few lines.
Response: According to the reviewer’s suggestion, we have explained this model in lines 134–139.